# Superficial Zoonotic Mycoses in Humans Associated with Cats

**DOI:** 10.3390/jof10040244

**Published:** 2024-03-24

**Authors:** Marcin Piorunek, Honorata Kubisiak-Rzepczyk, Aleksandra Dańczak-Pazdrowska, Tomasz Trafas, Jarosław Walkowiak

**Affiliations:** 1Veterinary Practice Marcin Piorunek, Olimpijska 12, 60-185 Skorzewo, Poland; 2Department of Dermatology, Poznan University of Medical Sciences, 60-355 Poznan, Poland; rzepczykh@ump.edu.pl (H.K.-R.); aleksandra.danczak-pazdrowska@ump.edu.pl (A.D.-P.); 3Department of Pulmonology, Allergology and Pulmonary Oncology, Poznan University of Medical Sciences, 60-569 Poznan, Poland; ttrafas@ump.edu.pl; 4Department of Pediatric Gastroenterology and Metabolic Diseases, Poznan University of Medical Sciences, 60-572 Poznan, Poland; jwalkowiak@ump.edu.pl

**Keywords:** superficial mycoses, zoonosis, epidemiology, cats, mycological examination

## Abstract

Dermatophytosis is a superficial fungal skin infection common in humans around the world and is one of the many zoonotic skin diseases that cat owners are at risk of contracting. This retrospective study was conducted based on a detailed analysis of the results of mycological examination and medical documentation of 56 patients diagnosed with cat-to-human dermatophytoses from January 2017 to July 2022. Zoonotic mycoses were diagnosed more frequently in young people and women. In children, lesions most often occurred in the scalp area, and in adults, in the glabrous skin area. Skin infections caused by *Microsporum canis* (*M. canis*) prevailed and were confirmed in 47 patients (83.9%). *Trichophyton mentagrophytes* (*T. mentagrophytes*) was found in nine (16.1%) patients. *M. canis* predominantly caused infections of the scalp, followed by lower limb infections. Hairy scalps were almost exclusively involved in children. The odds of diagnosing *M. canis* infection compared to *T. mentagrophytes* infection was significantly higher in the head than in other regions, especially among children. The positive predictive value of a direct macroscopic examination was relatively low.

## 1. Introduction

In 2022, the overall population of pet cats was estimated at 350 million and that of stray cats was estimated at 480 million in the world. The number of purebred and mixed-breed cats was 127 million that year, and they were the most frequently chosen pets in the European Union (EU). Germany and France are the two countries with the highest number of cats in the EU, at 15.200 and 14.900 million, respectively. Poland came in fourth place with an estimated 7.125 million cats. Compared with the cat population in Germany and France, countries such as Latvia, Ireland, and Estonia had a low number of household felines. However, the share of households with at least one cat in the European Union differs in 2022. Romania ranked the highest, with 48 percent of households with at least one cat, followed by Poland with 40 percent and Hungary with 35 percent of households with cats [1,2,3].

Dermatophytosis is a superficial dermatitis common worldwide and is one of the many zoonotic skin diseases that the owners of such a large cat population are at risk of contracting [4]. The infectious form of dermatophytes is arthrospora, which is formed by the fragmentation of fungal hyphae into small infectious spores. Skin damage is an important factor in the development of infection. Infectious arthrospores can survive in the environment for about a year [5].

The majority of keratinophilic infections in cats are caused by *M. canis*, *T. mentagrophytes*, and *Nannizzia gypsea* (*N. gypsea*), as well as other less common species. Some cats may be asymptomatic carriers of dermatophytes. Factors predisposing animals to the development of infection are improper hygiene, reduced immunity, and staying in large groups such as breeding farms and shelters [6]. The most common symptoms are circular alopecia, scaling, and erythematous margins around the central healing area. There may be papules, scales, scabs, erythema, clogging of hair follicles, discoloration, and changes in the appearance of nails. In selected cases, the lesions may be symmetrical; pruritus is generally absent or mild. Dermatophytosis in a cat most often appears on the face and ears and then spreads to the paws and other parts of the body (Figure 1).

The disease is transmitted through direct contact with an infected animal or element in the environment, as well as through parasites and in the form of an aerosol containing fungal spores. In most cats, hair loss is limited to specific areas of the skin. In the case of immunodeficiency, the disease may be multifocal or a generalized skin disease [5]. Dermatophytes are spread to humans mainly through contact with infected cats and contaminated household items or grooming accessories such as towels or rags, combs, brushes, clippers, transport cages, food bowls, litter boxes, cat toys, blankets, bedding, lab coats, leather gloves, mops, or even external parasites. Sofas, beds, chairs, and furniture contaminated with fungi may also be the cause of infection [5,7,8].

The most common zoophilic dermatophytes causing skin infections in humans include *M. canis*, *T. mentagrophytes*, and *Trichophyton verrucosum* (*T. verrucosum*) [9]. Cats are the main reservoir of *M. canis*, with some populations of these animals showing up to 100% infection rates [10]. Human-to-human infection with *M. canis* has been demonstrated; however, the ability to infect wanes after several transmissions [11].

The increased risk of infection is associated with inadequate animal care and non-compliance with hygiene rules by humans. Children are more at risk of infection because they are more likely to be in close contact with pets and follow hygiene rules less strictly [12]. Immune deficiencies, immunosuppression, and chronic antibiotic therapy, as well as unfavorable changes in the condition of the stratum corneum, favor the spread of dermatophytosis. Zoophilic dermatophytosis in children primarily occurs on the scalp and glabrous skin, and on glabrous skin in adults. On hairy skin, the changes appear as numerous inflammatory areas of baldness covered with scales, with the presence of evenly broken hair, or inflammatory nodules with pustules from which hair easily falls out with gentle pulling (Figure 2). In both cases, full hair regrowth occurs after recovery.

A typical presentation on glabrous skin is a circumferential erythematous lesion, often with a concentric arrangement, vesicles, papules, pustules, and scaling at the margins [13,14,15].

The aim of this study was to determine the etiology and location of zoonotic dermatophytoses in cat owners in Greater Poland.

## 2. Materials and Methods

This retrospective study was conducted on the basis of a detailed analysis of the results of the mycological examination and medical documentation of 56 patients diagnosed with cat-to-human dermatophytoses in the Medical Mycology Laboratory of the Department of Dermatology and Venereology at the Poznan University of Medical Sciences from January 2017 to July 2022. The number of direct microscopic examinations was 89 and was dependent on one or more locations of skin changes in individual patients.

Depending on the location of the skin lesions, mycological diagnosis included the following:–Epidermal scales taken with a scalpel or surgical spoon;–Hair collected with medical forceps from the middle parts of the lesion;–Swab material from the skin.

The next stage of the diagnosis was a direct microscopic examination to assess the elements of the fungal structure, such as hyphae and spores. The assessment of scales and hair changed by disease was performed by placing the test material on a basic slide, subjected to a solution of 40% dimethylsulfoxide with 10–20% KOH, and protected with a cover slide. The material obtained from the swab was stained using the Gram method or lactophenol cotton blue with methylene blue. The prepared slides were analyzed microscopically at 200–400-fold magnification (ZEISS Axio Lab.A1 Microscope and ZEISS Axiocam 105 color Microscope Camera, Carl Zeiss Microscopy GmbH, Jena, Germany). ZEN Microscopy Software 3.9 was used for microscopic imaging and data archiving (Carl Zeiss Microscopy GmbH, Jena, Germany). Fungi cultures were established to identify the species. As substrates for the cultivation of dermatophytes, an agar medium (SDA, Sabouraud Dextrose Agar, Pol-Aura, Zawroty, Poland) and actidione SDA were used. The cultivation of dermatophyte fungi was carried out for a minimum of 3 weeks. Species identification of dermatophytes grown on media was made on the basis of macroscopic and microscopic features visible in the obtained cultures (characteristic hyphae and spores).

Mycological diagnostics was carried out in accordance with the standards applicable in the Medical Mycology Laboratory at the Poznan University of Medical Sciences. All data were collected anonymously using electronic forms in the internal Redcap database https://redcap.ump.edu.pl/ (accessed on 1 January 2023). The study protocol was approved by the Bioethical Committee at the Poznan University of Medical Sciences. This study did not bear the characteristics of a medical experiment and was conducted in accordance with the Helsinki Declaration.

The results are presented as descriptive statistics (ranges and percentages). The odds ratio (OR) with a 95% confidence interval (CI) was calculated to compare the prevalence of dermatomycoses at different locations depending on causative factors. Statistical significance was set at *p* < 0.05.

## 3. Results

The study included 56 people, owners of pet cats, in the age range of 2 to 63 years. The median age of all the patients was 14.5 years old. The large group of patients included children up to 14 years of age. Females predominated, with 42 of the patients being females (75%). In adulthood, dermatophytosis was observed most frequently between 26 and 32 years of age (Figure 3).

In order to identify the dermatophyte species, diagnostic material from all patients was cultured, regardless of the direct examination result. In each case, a positive result was obtained. Skin infections caused by *M. canis* prevailed and were confirmed in 47 patients (83.9%) (Figure 4). *T. mentagrophytes* was diagnosed in nine patients.

Skin lesions covered various parts of the body. The most common localization of dermatophytosis was the head, followed by the lower limb, upper limb, trunk, and neck. *T. mentagrophytes* was found in sequence within the skin of the trunk, neck, upper limb, lower limb, and head (Figure 5).

A detailed analysis made it possible to determine the occurrence of particular species of fungi in relation to the individual parts of the body. In 32 patients, dermatophytosis was located in only one part of the body. In the remaining cases, skin lesions covered two or three body areas. The hairy scalp was a frequent site of changes, which was observed in 14 patients, of whom 13 were children, and *M. canis* infection was confirmed in each case. *M. canis* infection was also most common in other parts of the body, except for the back, where *T. mentagrophytes* infection was dominant (Table 1).

The results obtained by other authors regarding the localization of dermatophytoses are presented in Table 2.

*M. canis* was the most commonly detected causative factor for all body regions in the analyzed group of patients (Table 1). In the case of the location of the changes on the head, as opposed to other regions of the body, the odds of *M. canis* detection compared to *T. mentagrophytes* tended to be significant. The above observations apply to the entire study group and the population of children (Table 3).

Direct microscopic examination allowed the identification of the elements of the fungal structure (hyphae or/and spores) in 34 patients (60.7%). In the case of *M. canis*, the result was positive in 24 patients and negative in 23 patients; for *T. mentagrophytes*, the result was positive in 6 patients and negative in 3 patients (Figure 6).

## 4. Discussion

Pets serve valuable social roles in society. Cats reduce the level of stress and feelings of loneliness and help to participate in the socialization process [23,24]. Despite these undoubted benefits, cats pose a risk of zoonotic dermatophytoses in their owners, especially those with reduced immunity [25]. Persian, Himalayan, and other long-haired cats are more frequent sources of infection compared to European shorthair cats [26]. Cats staying outdoors, especially in rural areas, may be infected with the geophilic dermatophyte *N. gypsea*. Individuals exposed to contact with rodents are predisposed to *T. mentagrophytes* and *Trichophyton quinckeanum* (*T. quinckeanum*) infections, whereas those exposed to cattle are more likely to be infected with *T. verrucosum* [27]. Our research was retrospective and did not include data on the age and breed of the cats, place of residence, outdoor activities, and contacts with other animals. The above-mentioned factors could have been directly related to infection with a specific fungal species examined by us in cat owners.

Dermatophytes are the most common and highly infectious fungal pathogens in the world, responsible for most superficial skin and nail infections [28]. The prevalence of *M. canis* is highly variable and depends on various factors, such as geographic region, the population sampled, whether or not culture status is correlated with disease, and criteria for data collection and reporting [4]. Studies conducted in central Italy have shown that *M. canis* is the most common dermatophyte isolated from cats, whereas *T. mentagrophytes* is the most frequently recognized in humans [29]. A similar research carried out in the southern part of Italy has confirmed that *M. canis* is the most common dermatophyte, followed by *N. gypsea* and *Trichophyton terrestre* (*T. terrestre*) [10]. A survey conducted on stray cats in the Lisbon Metropolitan Area showed species with decreasing incidence: *M. canis*, *T. mentagrophytes*, and *T. verrucosum* [30]. A study conducted in two different geographic regions in the United States resulted in the identification of five different species of fungi, like *M. canis*, *T. verrucosum*, *T. mentagrophytes*, *Trichophyton rubrum* (*T. rubrum*), and *Epidermophyton spp*. *M. canis* was isolated only from cats in the south, whereas *T. rubrum* was isolated more frequently from cats in the north [31].

The estimated risk of developing dermatophytosis throughout human life is 10–20% [32]. Given the prevalence of dermatophytes and the high rates of infection, proper diagnosis and treatment are extremely important [33]. The study population was dominated by females. In the group of people under 18, females accounted for 37.5%, in the group of adults, the percentage was 65.7%. Children are more vulnerable to dermatophytosis because they are in direct contact with cats more often [12]. The risk of human infection with a dermatophyte from a cat results from the complex mutual relationships between them. It depends on the genetic factors that determine the animal’s behavior and the age, breed, hair length, and domestication of the cat. On the other hand, it is necessary to take into account human behavior towards a cat, which varies depending on the age and gender of the family member. It has been shown that women are more likely to verbally interact with a cat by sitting or lying on the floor, whereas men prefer to sit, for example, in an armchair and talk to the cat. Children most often approach the cat directly and stay in contact with it longer. The presence of more frequent dermatophyte infections observed on the head and neck is due to the fact that young cats are cuddled in these places, especially by small children, but also by adults. Older cats rub against their legs, are held in arms or placed on the chest and belly, or choose the place of direct contact with humans [34,35].

In this study, the most likely cause of infection was daily contact with a cat. Only three patients were infected by a family member who had a cat and lived elsewhere. Direct contact with an infected cat is the main route of dermatophyte transmission to humans [36]. The most frequently diagnosed dermatophyte in the study group was *M. canis*, the presence of which was confirmed in 47 cases. *T. mentagrophytes* was diagnosed in nine patients. Dermatophytosis, caused by *M. canis*, is the most common fungal infection in cats worldwide, and it is one of the most important infectious skin diseases in this species. The higher incidence of *M. canis* dermatophytosis in humans is due to the fact that it is the most common fungal infection in cats worldwide. Asymptomatic carriers and cats with sub-clinical and clinical signs of the disease may be sources of infection [6]. In the study group, the most common site of *M. canis* infection was the hairy scalp, confirmed in 14 patients (25%). Mycoses in this location were diagnosed in 13 children (93%), aged 2–13 years. In the remaining cases, *M. canis* included other parts of the body (in five patients, together with the scalp). The results of other studies indicate that tinea capitis caused by *M. canis* mainly affects children aged 3–7 years and it is less common in adults [37,38]. In the group of adult patients, the prevalence varies from 1.5 to 44.3% depending on the country [22]. In the study group, *T. mentagrophytes* infection was diagnosed in nine patients and included upper and lower limbs, neck, chest, back, stomach, and nose, but not in hairy scalps in any case. These observations are consistent with reports by other authors [39]. Diagnosis of zoonotic dermatophytes is based on information from the history, physical examination, direct microscopic examination, culture, and sequencing of the internal transcribed spacer (ITS). Microscopy is a relatively quick diagnostic method as the sample can be assessed immediately, less than 1 h after collection, and the false-negative rate ranges from 5 to 15% in clinical practice [10,40]. Treatment of localized dermatophytosis on the skin of various parts of the body is usually effective and undertaken more quickly by family physicians using topical medications. Systemic therapy is used to treat extensive skin lesions. Infections located on the scalp constitute a significant diagnostic problem and require specialized dermatological diagnosis, including clinical assessment and mycological tests, and usually general treatment [41]. Our experience shows that in the early stages, fungal lesions on the hairy skin may not be visible and may have few symptoms. Older children are often ashamed to show their parents skin changes, which delays diagnosis and appropriate treatment. Patients included in the study, after diagnostics performed at the Medical Mycology Laboratory, underwent treatment in various dermatological centers in Greater Poland.

## 5. Conclusions

Zoonotic mycoses are diagnosed more frequently in young people and women. In children, lesions most often occur in the scalp area, and in adults, in the glabrous skin area. The predominant etiological factor is *M. canis*, which predominantly causes infections of the scalp, followed by trunk infections. Hairy scalps are almost exclusively involved in children. The odds of diagnosing *M. canis* infection compared to *T. mentagrophytes* infection are significantly higher in the head than in other regions, especially among children. The positive predictive value of a direct macroscopic examination is relatively low.

## Figures and Tables

**Figure 1 jof-10-00244-f001:**
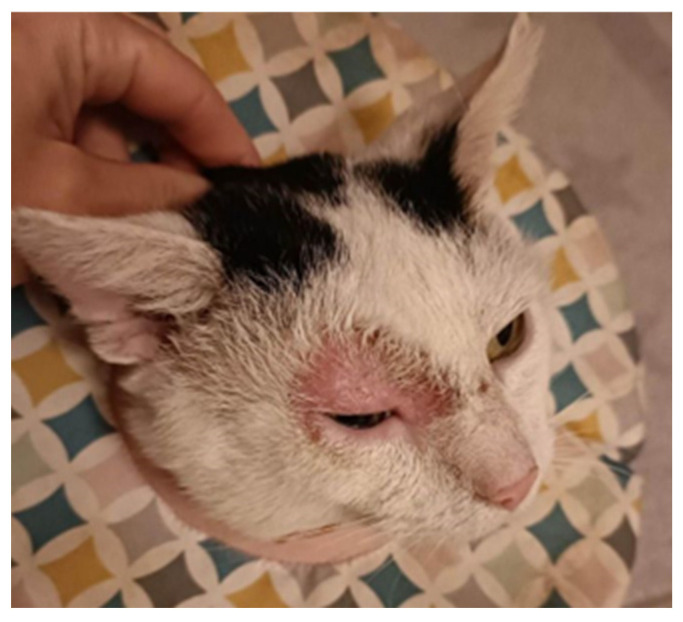
Skin changes in the course of dermatophytosis observed in cats (courtesy of Agnieszka Wolska).

**Figure 2 jof-10-00244-f002:**
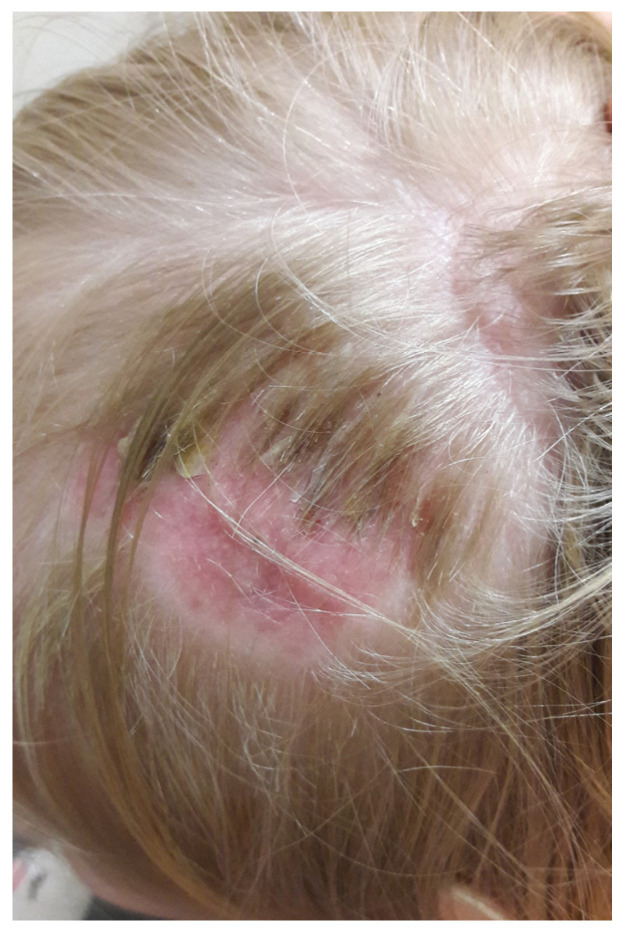
Circular alopecia visible on the hairy scalp of a child (courtesy of Honorata Kubisiak-Rzepczyk).

**Figure 3 jof-10-00244-f003:**
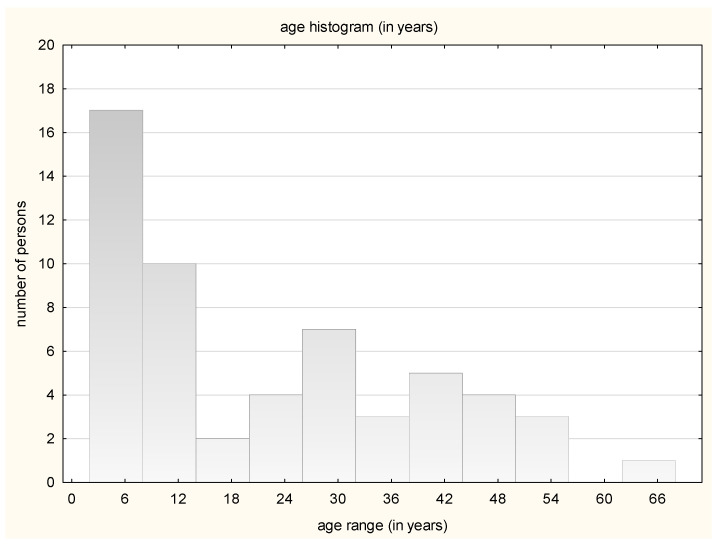
Number of patients in particular age groups.

**Figure 4 jof-10-00244-f004:**
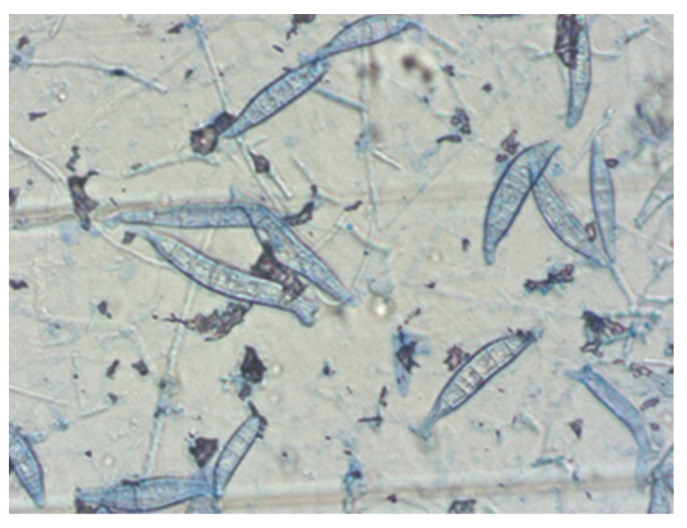
*M. canis* in microscopic examination (magnification ×200) (courtesy of Honorata Kubisiak-Rzepczyk).

**Figure 5 jof-10-00244-f005:**
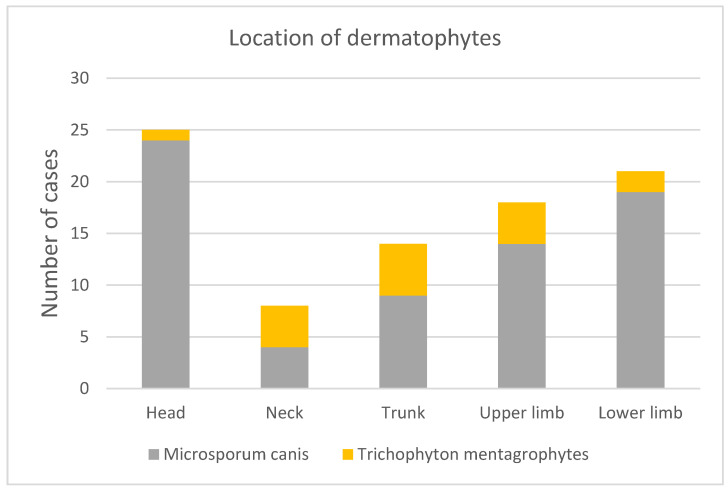
Localization of skin lesions caused by particular dermatophytes.

**Figure 6 jof-10-00244-f006:**
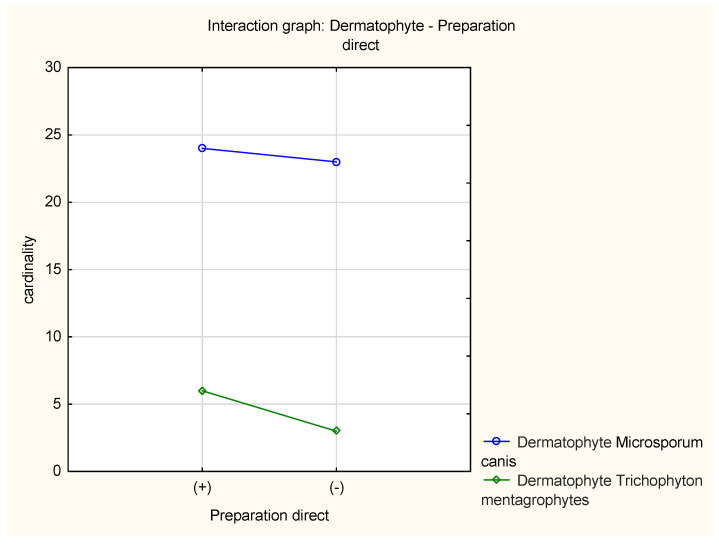
Direct microscopic examination results: positive result (+); negative result (−).

**Table 1 jof-10-00244-t001:** The occurrence of specific dermatophytes in relation to particular body regions.

General Location	Detailed Location	Number of Cases [%]		Number of Dermatophyte Species
Head	Hairy scalp	14	[16.28]	*Microsporum canis*—14
	Forehead	4	[4.65]	*Microsporum canis*—4
	Nose	1	[1.16]	*Trichophyton mentagrophytes*—1
	Cheeks	5	[5.81]	*Microsporum canis*—5
	Ear	1	[1.16]	*Microsporum canis*—1
Neck	Neck	8	[9.30]	*Microsporum canis*—4
				*Trichophyton mentagrophytes*—4
Trunk	Chest	5	[5.81]	*Microsporum canis*—3
				*Trichophyton mentagrophytes*—2
	Back	3	[3.49]	*Microsporum canis*—1
				*Trichophyton mentagrophytes*—2
	Stomach	6	[6.98]	*Microsporum canis*—5
				*Trichophyton mentagrophytes*—1
Upper limb	Palm	1	[1.16]	*Trichophyton mentagrophytes*—1
	Wrist	1	[1.16]	*Microsporum canis*—1
	Forearm	7	[8.14]	*Microsporum canis*—4
				*Trichophyton mentagrophytes*—3
	Arm	2	[2.33]	*Microsporum canis*—2
	Arms and forearms	7	[8.14]	*Microsporum canis*—7
Lower limb	Shank	3	[3.49]	*Microsporum canis*—2
				*Trichophyton mentagrophytes*—1
	Thigh	10	[11.63]	*Microsporum canis*—9
				*Trichophyton mentagrophytes*—1
	Thighs and shanks	7	[8.14]	*Microsporum canis*—7
	Buttock	1	[1.16]	*Microsporum canis*—1

**Table 2 jof-10-00244-t002:** The occurrence of specific dermatophytes in particular body regions—chosen articles.

Author	Localization	Number of Cases		Dermatophyte
Alteras et al. Mycopathologia. 1981 [16]	*tinea capitis*	3	[27.27%]	*Microsporum canis*
	*tinea faciei*	3	[27.27%]	
	*tinea barbae*	3	[27.27%]	
	*tinea manuum*	1	[9.09%]	
	*auricular lobe*	1	[9.09%]	
Asticcioli et al. New Microbiol. 2008 [17]	*tinea capitis*	13	[92.8%]	*Microsporum canis*
	*tinea corporis*	17	[65.4%]	*Microsporum canis*
		3	[11.5%]	*Trichophyton mentagrophytes*
	*tinea pedis*	8	[32%]	*Trichophyton mentagrophytes*
	*tinea unguium*	3	[13.6%]	*Trichophyton mentagrophytes*
Brosh-Nissimov et al. Mycoses. 2018 [11]	*tinea corporis*	53	[10.56%]	*Microsporum canis*
Hiruma et al. Dermatologica. 1988 [18]	*tinea faciei*	1		*Microsporum canis*
Kallel et al. Mycol Med. 2017 [19]	*tinea capitis*	634	[67.00%]	*Microsporum canis*
		6	[0.66%]	*Trichophyton mentagrophytes*
Segundo et al. Rev Iberoam Micol. 2004 [20]	*tinea capitis*	21	[46.65%]	*Microsporum canis*
	*tinea faciei*	1	[2.17%]	
	*tinea corporis*	17	[36.96%]	
	*tinea pedis*	5	[10.87%]	
	*tinea unguium*	2	[4.35%]	
Watanabe et al. Med Mycol J. 2022 [21]	*tinea capitis*	2		*Microsporum canis*
Yang et al. Mycopathologia. 2021 [22]	*tinea capitis* *tinea corporis* *tinea pedis* *tinea unguium*	1		*Microsporum canis*

**Table 3 jof-10-00244-t003:** The odds of diagnosing *M. canis* infection compared to *T. mentagrophytes* depending on the region of the body.

**Study Group**	**OR (95%CI)**	** *p* **
Head vs. other regions	7.1489 (0.886–57.654)	0.03238
Head vs. trunk	9.60 (0.951–96.922)	0.02760
**Pediatric Population**	**OR (95%CI)**	** *p* **
Head vs. other regions	15.6199 (6.7981–184.7301)	<0.0001
Head vs. trunk	25.3756 (5.9906–95.8176)	<0.0001

OR—odds ratio; 95% CI—95% confidence intervals.

## Data Availability

Data are contained within the article.

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
