# Peer review of "Superficial Zoonotic Mycoses in Humans Associated with Cats"

_jof, 2024, doi:10.3390/jof10040244_

Round 1
Reviewer 1 Report
Comments and Suggestions for Authors
The study is a good contribution to the knowledge of the fungi that cause dermatophytosis, however, I consider that it can be improved. My suggestions are as follows:
1. I would expect to find information about zoonotic mycoses associated with different animals, for example, cats, dogs, horses, rabbits, guinea pigs, even hedgehogs, however, the study only refers to cats. I suggest to adjust the title.
2. Lines 43-44 and 125. It is not necessary to write the abbreviation of the species in parentheses. The first time they appear in the text the scientific names are written in full and thereafter they are written in abbreviated form.
3. Line 44. Has the taxonomy of Microsporum spp. been revised? Is M. gypseum still part of the Microsporum complex?
4. Line 54. What elements of the environment can transmit the infection? I suggest specifying.
5. Lines 64 to 66. The author’s state: "The increased risk of infection is associated with inadequate animal care and noncompliance with hygiene rules by humans" and immediately after indicate: "In order to reduce the risk of infection, it is important to ensure proper animal care and human hygiene." Both lines seem redundant, I suggest rewriting the sentence.
6. Line 78. I suggest specifying: Cat owners from which region? does cat-associated dermatophytosis have the same etiology worldwide?
7. Discussion: It seems to me that the discussion can be further enriched, because, as it is presented, some doubts arise, for example:
· Particularly lines 173-194, practically only present a summary of the results obtained and mention that it is consistent with other studies, but do not discuss anything else Why did they obtain those results?
· Why in children is it more common in the head and in adults in the trunk? Is this particularly due to the age of the host, the species of fungus or is it more associated to the behavior of each cat? Those of us who have had many cats know that each one has preferences for the places where to sleep or cuddle, there are some who prefer to be near the head or neck, others who enjoy being on the chest or legs and others who like to be hugged by their owners. Even this varies depending on the age of the cat, when they are puppies, they tend to cuddle close to the neck and when they mature, they may modify that behavior.
· Why is the fact that it occurs on different parts of the body depending on the age of the patient important for the knowledge, treatment or prevention of cat-associated dermatophytosis?
Author Response
Response sheet
Manuscript ID: jof-2876316
“Superficial Zoonotic Mycoses in Humans” by Marcin Piorunek, Honorata Kubisiak-Rzepczyk, Aleksandra DaÅ„czak-Pazdrowska, Tomasz Trafas, JarosÅ‚aw Walkowiak
Dear, Mr. Esber Dai
Managing Editor
Journal of Fungi
We appreciate the opportunity to submit our manuscript entitled “Superficial Zoonotic Mycoses in Humans” to Journal of Fungi.
Thank you very much for your comments and suggestions that help considerably to improve the manuscript and increase its originality. We have made the revision of the submitted article. Changes are marked in red. Below point-to-point response to reviewers comments.
Responses and revision notes:
Response to Reviewer 1 Comments
The study is a good contribution to the knowledge of the fungi that cause dermatophytosis, however, I consider that it can be improved. My suggestions are as follows:
Point. 1: I would expect to find information about zoonotic mycoses associated with different animals, for example, cats, dogs, horses, rabbits, guinea pigs, even hedgehogs, however, the study only refers to cats. I suggest to adjust the title.
Thank you very much for your valuable comment. The authors' intention was to describe zoonotic mycoses associated only with cats. We intend to develop research material on zoonotic mycoses associated with other pets and farm animals in the future. As suggested, the title has been changed for “Superficial zoonotic mycoses in humans associated with cats”.
Point. 2: Lines 43-44 and 125. It is not necessary to write the abbreviation of the species in parentheses. The first time they appear in the text the scientific names are written in full and thereafter they are written in abbreviated form.
Thank you very much for your above suggestion. The scientific names are written in abbreviated form: M. canis, T. mentagrophytes, M. distortum.
Point. 3: Line 44. Has the taxonomy of Microsporum spp. been revised? Is M. gypseum still part of the Microsporum complex?
Thank you very much for your suggestion. A sentence in the text: "The majority of keratinophilic infections in cats are caused by M. canis, T. mentagrophytes and Microsporum gypseum as well as other less common species" was changed. The name Microsporum gypseum (former name) was changed to Nannizzia gypsea (current name). Nannizzia gypsea belongs to the Nannizzia complex.
Point. 4: Line 54. What elements of the environment can transmit the infection? I suggest specifying.
Thank you very much for your above suggestion. The sentence in the text has been supplemented with additional information. Dermatophytes are spread to humans mainly through contact with infected cats and contaminated household items or grooming accessories such as towels or rags, combs, brushes, clippers, transport cages, food bowls, litter boxes, cat toys, blankets, bedding and also lab coats, leather gloves, mops or even external parasites. Sofas, beds, chairs and furniture contaminated with fungi may also be the cause of infection. [Newbury S, http://oacu.od.nih.gov/disaster/ShelterGuide.pdf,], [ [https://www.americanhumane.org, 2010], [Moriello KA, 2017].
Point. 5: Lines 64 to 66. The author’s state: "The increased risk of infection is associated with inadequate animal care and noncompliance with hygiene rules by humans" and immediately after indicate: "In order to reduce the risk of infection, it is important to ensure proper animal care and human hygiene." Both lines seem redundant, I suggest rewriting the sentence.
Thank you very much for your suggestion. The following sentence has been removed from the text: In order to reduce the risk of infection, it is important to ensure proper animal care and human hygiene.
Point. 6: Line 78. I suggest specifying: Cat owners from which region?
Thank you very much for your above suggestion. The aim of the study was more precisely defined: The aim of the study was to determine the etiology and location of zoonotic dermatophytoses in cat owners in Greater Poland (part of west side of Poland).
Does cat-associated dermatophytosis have the same etiology worldwide?
The following text was posted in the discussion:
The prevalence of M. canis is highly variable and depends on various factors such as geographic region, the population sampled, whether or not culture status is correlated with disease, and criteria for data collection and reporting [Moriello J, 2014]. Studies conducted in central Italy have shown, that M. canis was the most common dermatophyte isolated from cats, while T. mentagrophytes was the most frequently recognized in humans [Iorio R, 2007;]. A similar research carried out in the southern part of Italy confirmed that M. canis was the most common dermatophyte, followed by N. gypsea and Trichophyton terrestre (T. terrestre) [Cafarchia, 2004]. Survey completed in stray cats at the Lisbon Metropolitan Area showed species in decreasing incidence: M. canis, T. mentagrophytes and T. verrucosum [Duarte A, 2010]. A study conducted in two different geographic regions in the United States allowed identification of five different species of fungi, like M. canis, T. verrucosum, T. mentagrophytes, Trichophyton rubrum (T. rubrum) and Epidermophyton spp. M. canis was isolated only from cats in the south, while T. rubrum was recognized more frequently from cats in the north [Moriello, 1994].
Point. 7: Discussion: It seems to me that the discussion can be further enriched, because, as it is presented, some doubts arise, for example:
Point. 7a: Particularly lines 173-194, practically only present a summary of the results obtained and mention that it is consistent with other studies, but do not discuss anything else Why did they obtain those results?
Thank you very much for your valuable comment and suggestions. Discussion was supplemented with the following contents:
Table No. 2 (chapter - Results) shows the distribution of dermatophytoses confirmed by the other authors.
Persian, Himalayan and other long-haired cats are more frequent source of infection, compared to European shorthair cats. [Westhoff D.K, 2010]. Cats staying outdoors, especially in rural areas, may be infected with the geophilic dermatophyte N. gypsea. Individuals exposed to contact with rodents are predisposed to T. mentagrophytes and Trichophyton quinckeanum (T. quinckeanum) infections, while those exposed to cattle are more likely to be infected with T. verrucosum [T. Frymus, 2017]. Our research was retrospective and did not include data on the age and breed of the cats, place of residence and outdoor activities, or contacts with other animals. The above-mentioned factors could have directly related to infection with a specific fungus species in examined by us cat owners.
The following sentence has been removed: A similar relationship probably applies to women.
Point. 7b: Why in children is it more common in the head and in adults in the trunk? Is this particularly due to the age of the host, the species of fungus or is it more associated to the behavior of each cat? Those of us who have had many cats know that each one has preferences for the places where to sleep or cuddle, there are some who prefer to be near the head or neck, others who enjoy being on the chest or legs and others who like to be hugged by their owners. Even this varies depending on the age of the cat, when they are puppies, they tend to cuddle close to the neck and when they mature, they may modify that behavior.
Thank you very much for your above valuable suggestion. The following information has been added to the discussion:
The risk of human infection with a dermatophyte from a cat results from the complex mutual relations between them. It depends on the genetic factors that determine the animal's behavior and the age, breed, hair length and domestication of the cat. On the other hand, it is necessary to take into account human behavior towards a cat, which varies depending on the age and gender of the family member. It has been shown that women are more likely to verbally interact with a cat by sitting or lying on the floor, while men prefer to sit, for example, in an armchair and talk to the cat. Children most often approach the cat directly and stay in contact with it longer. The presence of more frequent dermatophyte infections observed on the head and neck is due to the fact that young cats are cuddled in these places, especially by small children, but also by adults. Older cats rub against legs, are held in arms or placed on the chest and belly, or choose the place of direct contact with humans [Turner DC, 2021], [Bradshaw].
Point. 7c: Why is the fact that it occurs on different parts of the body depending on the age of the patient important for the knowledge, treatment or prevention of cat-associated dermatophytosis?
Thank you very much for your valuable comment. The discussion has been supplemented with the following text:
Treatment of localized dermatophytosis on the skin of various parts of the body is usually effective and undertaken more quickly by family physicians using topical medications. Systemic therapy is used to treat extensive skin lesions. Infections located on the scalp constitute a significant diagnostic problem and require specialized dermatological diagnosis and usually general treatment [Kaul S, 2017]. Our experience shows that the early stages, fungal lesions on the hairy skin may not be visible and may have few symptoms. Older children are often ashamed to show their parents skin changes, which delays diagnosis and appropriate treatment.
The manuscript has been linguistically proofread by an English translator.
The e-mail address of author Tomasz Trafas has been changed to ttrafas@ump.edu.pl
Sincerely yours,
Marcin Piorunek, DVM
e-mail: piorun.mp@gmail.com
Reviewer 2 Report
Comments and Suggestions for Authors
General comments
The manuscript by Piorunek et al is a well structured and presented concise review on the superficial zoonotic mycoses. The paper addresses this timely topic in an balanced manner, effectively conveying main relevant points to the readers. The manuscripts presents an in-depth coverage of the subject also with the help of several tables and figures. In general, I totally agree with presentation and the conclusions of this article, which are mainly in line with the current expert opinions and the current diagnostic approaches.
My comments/suggestions for improvement are listed below:
Specific comments
Major points
-
A figure (or two) showing typical clinical presentations in animals (cats) and humans with zoonotic dermatophytosis would certainly be of interest for the readership of the JoF and would „spice up“ this interesting manuscript. The new figures should be added to a revised manuscript.
-
Figure 2 – „Interaction Chart: Dermatophyte x Lesion Location“ – I think that a graphical depiction of the lesion localisation as columns or bars would be more appropriate.
-
Table 1 - It would be useful to add to the table summarizing the distribution of the skin lesions of dermatophytoses and of the specific dermatophytes the citations of the underlying studies. A referenced table should be part of the revised manuscript.
Minor points
-
Figure 2 – „Interaction Chart: Dermatophyte x Lesion Location“ – the name of the figure is somehow awkward and could be optimized. E.g., what exactly “interaction chart” means
Author Response
Response sheet
Manuscript ID: jof-2876316
“Superficial Zoonotic Mycoses in Humans” by Marcin Piorunek, Honorata Kubisiak-Rzepczyk, Aleksandra DaÅ„czak-Pazdrowska, Tomasz Trafas, JarosÅ‚aw Walkowiak
Dear, Mr. Esber Dai
Managing Editor
Journal of Fungi
We appreciate the opportunity to submit our manuscript entitled “Superficial Zoonotic Mycoses in Humans” to Journal of Fungi.
Thank you very much for your comments and suggestions that help considerably to improve the manuscript and increase its originality. We have made the revision of the submitted article. Changes are marked in red. Below point-to-point response to reviewers comments.
Responses and revision notes:
Response to Reviewer 2 Comments
The manuscript by Piorunek et al is a well structured and presented concise review on the superficial zoonotic mycoses. The paper addresses this timely topic in an balanced manner, effectively conveying main relevant points to the readers. The manuscripts presents an in-depth coverage of the subject also with the help of several tables and figures. In general, I totally agree with presentation and the conclusions of this article, which are mainly in line with the current expert opinions and the current diagnostic approaches.
Point. 1: A figure (or two) showing typical clinical presentations in animals (cats) and humans with zoonotic dermatophytosis would certainly be of interest for the readership of the JoF and would „spice up“ this interesting manuscript. The new figures should be added to a revised manuscript.
Thank you very much for your above suggestion. The following pictures are included in the manuscript:
Figure 1. Skin changes in the course of dermatophytosis observed in cat (courtesy of Agnieszka Wolska).
Figure 2. Circular alopecia visible on the hairy scalp in child (courtesy of Honorata Kubisiak-Rzepczyk).
Figure 4. Microsporum distortum in microscopic examination (courtesy of Honorata Kubisiak-Rzepczyk).
Point. 2: Figure 2 (now Figure 5) – „Interaction Chart: Dermatophyte x Lesion Location“ – I think that a graphical depiction of the lesion localization as columns or bars would be more appropriate.
Thank you very much for your suggestion. Graphical depiction of the skin lesion localization caused by particular dermatophytes as columns was applied - Figure 5.
Point. 3: Table 1 - It would be useful to add to the table summarizing the distribution of the skin lesions of dermatophytoses and of the specific dermatophytes the citations of the underlying studies. A referenced table should be part of the revised manuscript.
Table No. 2 (chapter - Results) shows the distribution of dermatophytoses confirmed by the other authors.
Author |
Localization |
Number of cases |
Dermatophyte |
|
Alteras et al. Mycopathologia. 1981 |
tinea capitis |
3 |
[27.27%] |
Microsporum canis |
tinea faciei |
3 |
[27.27%] |
||
tinea barbae |
3 |
[27.27%] |
||
tinea manuum |
1 |
[9.09%] |
||
auricular lobe |
1 |
[9.09%] |
||
Asticcioli et al. New Microbiol. 2008 |
tinea capitis |
13 |
[92.8%] |
Microsporum canis |
tinea corporis |
17 |
[65.4%] |
Microsporum canis |
|
3 |
[11.5%] |
Trichophyton mentagrophytes |
||
tinea pedis |
8 |
[32%] |
Trichophyton mentagrophytes |
|
tinea unguium |
3 |
[13.6%] |
Trichophyton mentagrophytes |
|
Brosh-Nissimov et al. Mycoses. 2018 |
tinea corporis |
53 |
[10.56] |
Microsporum canis |
Hiruma et al. Dermatologica. 1988 |
tinea faciei |
1 |
|
Microsporum canis |
Kallel et al. Mycol Med. 2017 |
tinea capitis |
634 |
[67.00%] |
Microsporum canis |
6 |
[0.66%] |
Trichophyton mentagrophytes |
||
Segundo et al. Rev Iberoam Micol. 2004 |
tinea capitis |
21 |
[46.65%] |
Microsporum canis |
tinea faciei |
1 |
[2.17%] |
||
tinea corporis |
17 |
[36.96%] |
||
tinea pedis |
5 |
[10.87%] |
||
tinea unguium |
2 |
[4.35%] |
||
Watanabe et al. Med Mycol J. 2022 |
tinea capitis |
2 |
|
Microsporum canis |
Yang et al. Mycopathologia. 2021 |
tinea capitis tinea corporis tinea pedis tinea unguium |
1 |
|
Microsporum canis |
Point. 4: Figure 2 – „Interaction Chart: Dermatophyte x Lesion Location“ – the name of the figure is somehow awkward and could be optimized. E.g., what exactly “interaction chart” means.
Thank you very much for your valuable comment. The name of the figure 2 „Interaction Chart: Dermatophyte x Lesion Location” was changed for “Location of dermatophytes”
The following sentence has been removed: A similar relationship probably applies to women.
The manuscript has been linguistically proofread by an English translator.
The e-mail address of author Tomasz Trafas has been changed to ttrafas@ump.edu.pl
Sincerely yours,
Marcin Piorunek, DVM
e-mail: piorun.mp@gmail.com
Round 2
Reviewer 1 Report
Comments and Suggestions for Authors
I have reviewed the latest version of the manuscript on superficial mycoses submitted by the authors and consider it suitable for publication with the modifications made.
It seems to me that the added information has been a considerable improvement, particularly in discussion.
Author Response
Dear Sir,
Thank you very much for your previous valuable comments and suggestions that help considerably to improve the manuscript and increase its originality.
Sincerely yours,
Marcin Piorunek, DVM
e-mail: piorun.mp@gmail.com